

# Precise dating of deglacial Laptev Sea sediments via [14]C and authigenic [10]Be/[9]Be – assessing local [14]C reservoir ages

Arnaud Nicolas[1,2], Gesine Mollenhauer[1,2,3], Johannes Lachner[4], Konstanze Stübner[4], Maylin Malter[1], Jutta Wollenburg[1], Hendrik Grotheer[1,3], Florian Adolphi[1,2]

[1]Alfred Wegener Institute, Bremerhaven, Germany
[2]Department of Geosciences, University of Bremen, Bremen, Germany
[3]MARUM-Center for Marine Environmental Sciences, University of Bremen, Bremen, Germany
[4]Helmholz-Zentrum Dresden-Rossendorf, Dresden, Germany

Correspondence: Arnaud Nicolas (arnaud.nicolas@awi.de) and Florian Adolphi (florian.adolphi@awi.de)

**Abstract**

Establishing accurate chronological frameworks is imperative for reliably identifying lead-lag dynamics within the climate system and enabling meaningful inter-comparisons across diverse paleoclimate proxy records over long time periods. Robust age models provide a solid temporal foundation for establishing correlations between paleoclimate records. One of the primary challenges in constructing reliable radiocarbon-based chronologies in the marine environment is to determine the regional marine radiocarbon reservoir age correction. Calculations of the local marine reservoir effect (ΔR) can be acquired using [14]C-independent dating methods, such as synchronization with other well-dated archives. The cosmogenic radionuclide [10]Be offers such a synchronization tool. Its atmospheric production rate is controlled by the global changes in the cosmic ray influx, caused by variations in solar activity and geomagnetic field strength. The resulting fluctuations in the meteoric deposition of [10]Be are preserved in sediments and ice cores and can thus be utilized for their synchronization. In this study, for the first time, we use the authigenic [10]Be/[9]Be record of a Laptev Sea sediment core for the period 8-14 kyr BP and synchronize it with the [10]Be records from absolutely dated ice cores. Based on the resulting absolute chronology, a benthic ΔR value of +345 ± 60 [14]C years was estimated for the Laptev Sea, which corresponds to a marine reservoir age of 848 ± 90 [14]C years. The ΔR value was used to refine the age-depth model for core PS2458-4, establishing it as a potential reference chronology for the Laptev Sea. We also compare the calculated ΔR value with modern estimates from the literature and discuss its implications for the age-depth model.

## 1 Introduction

Paleoclimate reconstructions can provide useful information about the dynamics of the climate system under different boundary conditions. Investigating how the climate variations propagate in space and time can provide important information about the underlying driving mechanisms (Adolphi et al., 2018; Czymzik et al., 2016b, a; Reinig et al., 2021). To correctly assess regional variations and spatio-temporal patterns in climate fluctuations, it is crucial to construct precise chronological frameworks. These frameworks serve as the temporal backbone for establishing correlations between paleoclimate records derived from marine, terrestrial, and ice-core archives. However, uncertainties in chronologies across different paleoclimate records often hinder the precise assessment of paleoclimate dynamics involving multiple records from different sites and archives (Southon, 2002).

One of the key challenges for constructing precise chronologies in the marine realm is to estimate the regional marine radiocarbon reservoir age correction, especially in polar regions (Alves et al., 2018; Heaton et al., 2023).



For constructing an age-depth model using $^{14}$C dates of marine samples, it is crucial to include a precise marine
reservoir age (MRA). The MRA is the radiocarbon age difference between a marine sample and its contemporary
atmosphere (Stuiver et al., 1986). According to the most recent radiocarbon calibration curve, Marine20, the global
average marine reservoir age is approximately 500 $^{14}$C years during the Holocene period (0 - 11.6 kyr BP) (Heaton
et al., 2020). However, regional differences in ocean-atmosphere exchange and internal ocean mixing can result
in large regional deviations from this global mean (Heaton et al., 2023). Therefore, the local marine reservoir
effect, ΔR was introduced and is defined as the difference between the regional and the modelled global marine
reservoir ages (Reimer and Reimer, 2001; Stuiver et al., 1986).

There is only one study that has provided modern MRA estimates for the Laptev Sea (Bauch et al., 2001). In this
study, the MRAs range from 295 ± 45 to 860 ± 55 $^{14}$C years, with a mean value of 451 ± 72 $^{14}$C years. Estimates
for MRA from the early deglaciation (~15 kyr BP) to the Holocene period for creating reliable deglacial
chronologies in the Laptev Sea are so far not available.

In order to provide estimates of the local ΔR back in time the samples must be independently dated by other means
than $^{14}$C. This can for example be achieved by synchronization to other well-dated archives. Cosmogenic
radionuclides such as $^{10}$Be and provide such a synchronization tool (Adolphi et al., 2018; Adolphi and Muscheler,
2016; Czymzik et al., 2018, 2020; Muscheler et al., 2014; Southon, 2002).

The cosmogenic radionuclides Beryllium-10 ($^{10}$Be, half-life = 1.387 ± 0.012 Myr) (Chmeleff et al., 2010;
Korschinek et al., 2010) and Carbon-14 ($^{14}$C, half-life = 5.700 ± 0.03 kyr) (Audi et al., 2003) are mainly produced
in Earth's upper atmosphere in a particle cascade that is triggered when galactic cosmic rays interact with atoms
in the atmosphere (Lal and Peters, 1967; Dunai and Lifton, 2014). The flux of these cosmic rays reaching Earth
is controlled by variations in the heliomagnetic and geomagnetic shielding (Lal and Peters, 1967; Masarik and
Beer, 1999) During periods of higher solar activity and/or geomagnetic field strength, production rates of $^{10}$Be
and $^{14}$C are decreased, whereas the production rates are higher during reduced solar activity and/or lower magnetic
field strength. The production rates of both cosmogenic radionuclide isotopes co-vary globally due to these
external processes.

Following production in the atmosphere, $^{14}$C oxidizes to $^{14}CO_2$, enters the global carbon cycle and is incorporated
in environmental archives such as tree-rings, foraminifera, or speleothems. Annually, gigatons of carbon are
exchanged between the Earth's active reservoirs of the atmosphere, biosphere and the ocean, within the global
carbon cycle. Carbon is recycled and reused within these reservoirs and some reservoirs such as the deep ocean
can take hundreds of years to recycle carbon back to the atmosphere. The resulting heterogenous distribution of
radiocarbon among the different reservoirs stress the importance to understand and determine precise reservoir
ages.

In the atmosphere, the production of $^{10}$Be in the more stably layered stratosphere is higher than in the troposphere.
About 63 % of $^{10}$Be is produced in the stratosphere, 30 % in the tropical and subtropical troposphere together and
7 % in the polar troposphere(Adolphi et al., 2023; Poluianov et al., 2016). $^{10}$Be is adsorbed onto aerosol particles,



mixed during about 1-yr residence time in the stratosphere, and is then transported and deposited on Earth's
surfaces through wet and dry deposition (Raisbeck et al., 1981; Zheng et al., 2023). The $^{10}$Be production rates are
highest in the high-latitude stratosphere due to the weaker shielding of the cosmic ray flux by the Earth's magnetic
field. However, the highest $^{10}$Be fluxes to Earth's surface are recorded in mid-latitudes because of the strong
regional exchange between stratosphere and troposphere and high precipitation rates leading to strong aerosol
scavenging (Heikkilä et al., 2013). Non-production processes such as variations in mixing, transport and
deposition of $^{10}$Be and $^{14}$C can complicate the reconstruction of cosmogenic radionuclide production rates from
paleoenvironmental archives. However, common variations in both cosmogenic radionuclide records are
considered to represent the cosmogenic radionuclide production signal, due to their common production
mechanism and different chemical behavior (Lal and Peters, 1967; Muscheler et al., 2008). $^{10}$Be production rate
changes are relatively well-known from independently dated ice-core records (Finkel and Nishiizumi, 1997; Yiou
et al., 1997), and this can serve as a synchronization target for other records of $^{10}$Be production rate changes.

In order to obtain reliable records of $^{10}$Be-production rate changes from marine sediments, the effects of variable
sedimentation rates and particle scavenging must be accounted for, which can be efficiently achieved by
measuring authigenic $^{10}$Be/$^{9}$Be (Bourles et al. 1989). The stable isotope $^{9}$Be is a trace component in all continental
rocks. It is released by weathering of silicate rocks and transported to the ocean mainly by rivers (von
Blanckenburg et al., 2015). $^{9}$Be (and to a lesser extent meteoric $^{10}$Be) is introduced into the ocean in its dissolved
form where it is mixed with dissolved $^{10}$Be of ocean water (mainly derived from atmospheric fallout, see above).
Since Be is particle reactive in seawater, dissolved $^{10}$Be/$^{9}$Be is incorporated in marine authigenic phases as
amorphous coating on sediment or it can be preserved in authigenic Fe-Mn oxyhydroxides (von Blanckenburg
and Bouchez, 2014). Therefore, in marine sediment the authigenic $^{10}$Be/$^{9}$Be ratio reflects the isotope ratio of
dissolved Be of the overlying water column at the time of sediment deposition (Bourles et al., 1989; von
Blanckenburg and Bouchez, 2014).

If the riverine input of $^{9}$Be remains relatively constant, $^{9}$Be and $^{10}$Be are well-mixed (i.e., at sites >200 km from
the coast) (Wittmann et al., 2017), and the mixing of prevalent water-masses does not change, then authigenic
$^{10}$Be/$^{9}$Be should primarily reflect changes in the cosmogenic production rates of $^{10}$Be. In the Arctic Ocean, the
spatial patterns of $^{10}$Be/$^{9}$Be in the water column are more heterogeneous than most other open ocean settings
because of the mixing of Atlantic waters with $^{10}$Be/$^{9}$Be values of 5 - 10 x $10^{-8}$ and Arctic Rivers with $^{10}$Be/$^{9}$Be
values of 0.3 - 1.5 x $10^{-8}$) (Frank et al., 2009).

The aim of this study is to explore the use of an authigenic $^{10}$Be/$^{9}$Be of a Laptev Sea sediment core for its
synchronization to $^{10}$Be-records from absolutely dated ice cores. Using this result, we aim to infer the the local
marine reservoir effect, ΔR for the Laptev Sea during the deglaciation. This is the first study to exploit variations
in $^{10}$Be production rates from Arctic marine sediments for stratigraphic purposes.




**2 Materials and methods**
**2.1 Sediment core location and initial chronology**
The sediment core PS2458-4 measured for $^9$Be and $^{10}$Be in this study, was retrieved in 1994 from the eastern
Laptev Sea continental margin (78°10.0′N, 133°23.9′E) at a water depth of 983 m (Fütterer, 1994) and
approximately about 518 km from the Lena Delta (Fig. 1). The 8 m long core consists of very dark olive-grey silty
clay of dominantly terrigenous origin (Fütterer, 1994). This core consists of a continuous high-sedimentation-rate
(77 cm /kyr) sequence representing the deglaciation period between approximately 16.5 and 9.3 kyr BP, followed
by a lower-sedimentation-rate (27 cm /kyr) early Holocene sequence (Fahl and Stein, 2012). A first chronology
of core PS2458-4 was established by accelerator mass spectrometry (AMS) $^{14}$C dating of calcareous foraminifera,
bivalves and wood samples for the sediment interval between 201 and 667 cm, corresponding to a time interval
between approximately 8.8 and 14.3 kilo-calendar years BP (kyr BP) (Spielhagen et al., 2005). To improve the
existing age-depth model, 7 new AMS $^{14}$C dates from mixed benthic foraminifera were used in combination with
7 $^{14}$C dates from mixed benthic foraminifera and bivalves from Spielhagen et al. (2005) and an initial age-model
was derived using OxCal4.4 (Ramsey, 2009) (see Table 2). The marine $^{14}$C dates were calibrated with the
Marine20 curve (Heaton et al., 2020). An average local marine reservoir effect (ΔR) value of -110 ± 28 $^{14}$C years
was used based on the nearest modern values from Bauch et al. (2001) available from the online database:
http://calib.org/marine/. This chronology provides the initial basis for the stratigraphic fine-tuning using $^{10}$Be/$^9$Be
as described below.

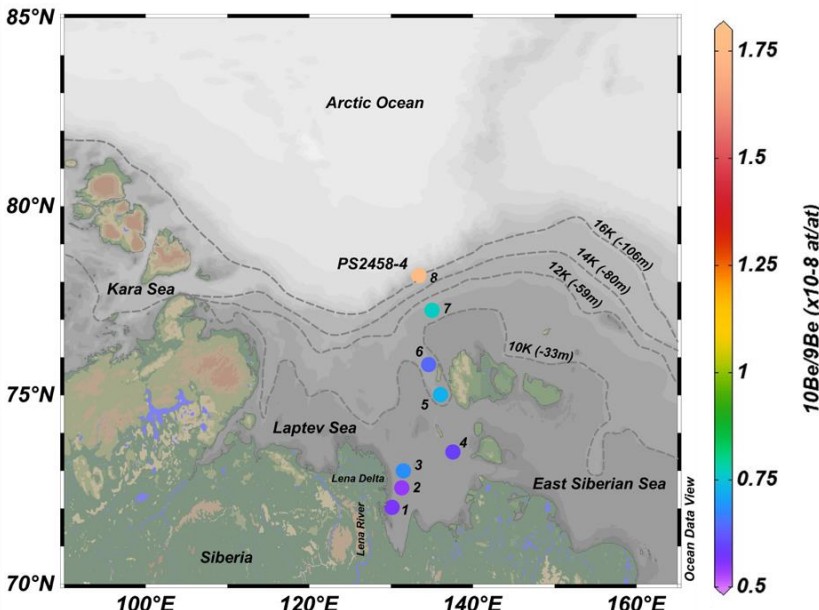

**Figure 1: Map of the Laptev Sea shelf showing the location of core PS2458-4 with core-top $^{10}$Be/$^9$Be concentration**
**(numbered colored circle 8) and $^{10}$Be/$^9$Be concentrations of modern surface sediments (numbered colored circles 1-7).**
**The dashed lines represent the reconstructed coastline extent at 4 different time periods (where 16K=16 kyr BP) with**
**corresponding water depth values in meters shown in brackets** (Klemann et al., 2015) **. The Map was created using**
**Ocean Data View** (Schlitzer, 2016)



### 2.2 Modern surface sediment samples from Laptev Sea

Seven modern surface sediment samples collected in the Laptev Sea were also included in the analysis (Figure 1, Table 1). Surface sediments with sample IDs 1 to 6 were collected during the Transdrift expeditions I and II in 1993 and 1994 using Van Veen grabs and large spade box corer (Kassens and Dmitrenko, 1995; Kassens and Karpiy, 1994). Sediment sample from core PS2728-2 with ID number 7 was recovered in 1995 with a large rectangular box sampler during the Arctic Expedition ARK-XI/1 (Rachor, 1997). The sediment samples used in this study are distributed along a transect from near to the Lena Delta towards the open ocean near the shelf break, close to where core PS2458-4 was retrieved.

### 2.3 Sample preparation and measurements

Fifty-four sediment samples were selected along core PS2458-4 and processed for Be isotope analysis at the Alfred Wegener Institute in Bremerhaven (Germany). According to the initial radiocarbon-based age model, the selected samples covered three large cosmogenic radionuclide production rate swings, as evidenced by ice core $^{10}$Be and tree-ring $^{14}$C records (e.g., Adolphi and Muscheler, 2016), that occurred between 8.5 and 11.5 kyr BP. The leaching of the authigenic Fe-Mn oxyhydroxides phase followed Gutjahr et al. (2007) with minor modifications. Sediment samples were freeze-dried, homogenised and ~1 g of sediment was treated with 1 M NaOAc and adjusted with HOAc to pH 4 to dissolve carbonates which were discarded. Subsequently, the sediments were leached using 0.04 M hydroxylamine ($NH_2OH$-$HCl$) in 15% HOAc at 95 °C for 4 h. We did not leach the exchangeable fraction as proposed by Gutjahr et al. (2007) as this contained less than 1 % of the Be leached in the hydroxylamine fraction with a very similar $^{10}$Be/$^9$Be ratio. An aliquot from the resulting leaching solution was sampled for stable $^9$Be measurements using an Atomic Emission Spectrophotometer at the Alfred Wegener Institute in Bremerhaven, Germany (Thermo Fisher Scientific Inc., ICP-OES-iCAP7400), with an internal Yttrium standard and standard addition. The remaining $^{10}$Be aliquot solution was spiked with a precisely weighed amount of $^9$Be-carrier (200, 300 or 500 µL of 1000 mg/L carrier solution, LGC 998969-73, $^{10}$Be/$^9$Be = $(3.74 \pm 0.31) \times 10^{-15}$ at/at) (Merchel et al., 2021). The purification of the samples largely followed the method outlined by Simon et al. (2016). The samples were evaporated, dissolved in distilled HCl and $NH_3$ was added for Be oxy-hydroxide precipitation from the solution at pH 8 - 9. The precipitate was recovered by centrifugation and then dissolved in 1 mL distilled 10.2 M HCl before loading onto a column filled with 15 mL Dowex® 1 x 8 (100-200 mesh) anion-exchange resin in order to remove Fe from the sample. Prior, the resin was rinsed with 20 mL MilliQ® water and conditioned with 30 mL 10.2 M HCl. The sample was then loaded onto the column and eluted using 30 mL 10.2 M HCl. A column filled with 10 mL 50 x 8 (100 - 200 mesh) cation-exchange resin was used to separate Be from B and Al. The resin was treated with 20 mL MilliQ® water followed by 20 mL 1 M HCl. The sample was loaded onto the column and the first 25 mL 1 M HCl eluent, which contain mainly B, were discarded. Be was eluted and collected with the next addition of 90 mL 1 M HCl. The resulting Be oxy-hydroxides were precipitated at pH 8 - 9 by addition of $NH_3$, then separated by centrifugation and washed 3 times by rinsing with MilliQ® water to remove all chlorides. The purified Be oxy-hydroxides were transferred into quartz vials, dried at 80 °C overnight and finally calcinated to BeO at 900 °C for 2 h. The BeO was mixed with Nb powder (Nb:BeO = 4 : 1 by weight) and pressed into a Cu cathode-holder for accelerator mass spectrometer (AMS) measurements. One blank and one replicate were measured with each batch of samples in order to assess reproducibility and background during the extraction procedure.



AMS measurements were performed at DREAMS (DREsden AMS) facility (Lachner et al., 2023; Rugel et al.,
2016). All measurements were done relative to the standard "SMD-Be-12" with a weighted mean value of (1.704
± 0.030) x $10^{-12}$ (Akhmadaliev et al., 2013). Authigenic $^{10}$Be/$^{9}$Be was calculated from the AMS results, the known
amount of carrier, and the measured authigenic $^{9}$Be-concentration from Inductively Coupled Plasma Atomic
Emission Spectroscopy (ICP-AES) (see Simon et al., 2016). Considering the recent age of the samples, we did
not correct for decay of $^{10}$Be. The correction would be in the order of 0.5 % and is an order of magnitude lower
than our combined measurement precision.
The preparation and measurement of the 7 new benthic foraminifera samples were undertaken based on the
standard operation procedures routinely used at the MICADAS $^{14}$C laboratory facility of the Alfred Wegener
Institute (Mollenhauer et al., 2021). Prior to measurement, care was taken to critically select appropriate and
sufficient number of foraminifera shells without brownish discolouration or authigenic calcite overgrowth to
reduce uncertainty in the radiocarbon dates (Wollenburg et al., 2023).
**2.4 Ice core $^{10}$Be record**
The ice core $^{10}$Be record used in this study (Fig. 2) consists of normalized, averaged values of two ice cores: the
West Antarctic Ice Sheet (WAIS) Divide ice core$^{10}$Be (Muschitiello et al., 2019; Sigl et al., 2016; Sinnl et al.,
2023) and the Greenland Ice Sheet Project Two (GISP2) $^{10}$Be fluxes  (Finkel and Nishiizumi, 1997). The ice core
fluxes had been corrected for climate influences by performing a regression against $\delta^{18}$O and snow accumulation
rates (Adolphi et al., 2018). Prior to averaging, each ice core had been transferred to the IntCal20 timescale using
the timescale transfer functions described in several previous studies (Adolphi and Muscheler, 2016; Adolphi et
al., 2018 and Sigl et al., 2016). The glacial section of WAIS had been synchronized to Greenland Ice-Core
Chronology 2005 (GICC05) by using volcanic (Svensson et al., 2020) and cosmogenic (Sinnl et al., 2023) tie
points. The data from each ice core were resampled (averaged) to 40-year resolution before stacking. In order to
facilitate a comparison between ice core and marine $^{10}$Be changes, we modelled the expected marine signal from
the ice core record following Christl (2007). We chose a 350-year residence time of Beryllium in the water column
prior to deposition as this leads to a good agreement of amplitudes of the modelled centennial changes in $^{10}$Be to
the measured $^{10}$Be/$^{9}$Be changes seen in the sediment. This 350-year residence time is within the range of values
(80 ± 5 to 500 ± 25 years) reported in Arctic Ocean calculated from sedimentary fluxes and inventories (Frank et
al., 2009).



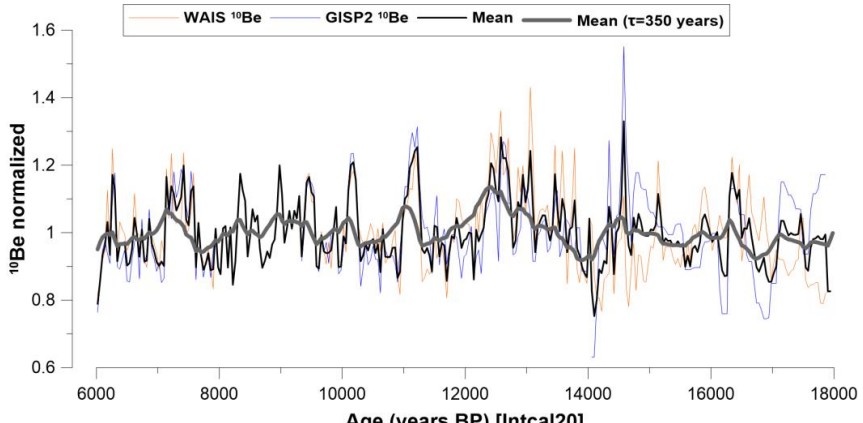


**Figure 2: WAIS (orange)** (Muschitiello et al., 2019; Sigl et al., 2016; Sinnl et al., 2023) **and GISP2 (blue)** (Finkel and Nishiizumi, 1997) $^{10}$**Be fluxes corrected for correlation to ice core accumulation rates and δ**$^{18}$**O, plotted on the IntCal20 timescale. The thick black line shows the mean of both datasets and the bold grey line depicts the modelled oceanic** $^{10}$**Be signal assuming a residence time (τ) of 350 years for** $^{10}$**Be in the water column.**

226

### 3 Results

The concentrations of $^9$Be, $^{10}$Be and $^{10}$Be/$^9$Be atomic ratios from core PS2458-4 are displayed in Fig. 3 and the data are shown in Table S2. The dominant feature is an increasing trend of $^{10}$Be/$^9$Be from the bottom to the top of the core. The modern surface sediment $^{10}$Be/$^9$Be values ([0.54 - 0.76] x $10^{-8}$) from the offshore transect spanning from the Lena Delta to the core site (Table 1, Fig. 1) are consistent with $^{10}$Be/$^9$Be of Lena water samples ([0.62 ± 0.07] x $10^{-8}$) (Frank et al., 2009) and within the same range as PS2458-4 $^{10}$Be/$^9$Be ([0.53 - 1.77] x $10^{-8}$). They show an increasing trend from the Lena Delta to the open ocean (Fig. 1). The modern values close to the Lena are consistent with the lowest $^{10}$Be/$^9$Be values of PS2458-4 during the deglaciation, when the core-site was proximal to the paleo-river mouth of the Lena (see Figure 1).

**Table 1. Information about location, water depth, distance from Lena Delta and concentration of authigenic**$^{10}$**Be,**$^9$**Be,**$^{10}$**Be/**$^9$**Be ratio leached of the modern surface sediment samples**.

| Sample name | Sample ID | Latitude (°) | Longitude (°) | Water Depth (m) | Approx. distance from Lena Delta (km) | $^9$Be (at/g) [x$10^{16}$] | $^{10}$Be (at/g) [x$10^8$] | $^{10}$Be/$^9$Be (at/at) [x$10^{-8}$] |
|---|---|---|---|---|---|---|---|---|
| IK93Z4-4 | 1 | 72.03 | 130.13 | 14 | 28 | 1.12 | 0.63 | 0.56 |
| IK9307-3 | 2 | 72.55 | 131.30 | 20.7 | 61 | 1.60 | 0.86 | 0.54 |
| IK9316-6 | 3 | 73.00 | 131.50 | 27.8 | 65 | 1.89 | 1.15 | 0.61 |
| IK9318-5 | 4 | 73.50 | 137.55 | 24 | 269 | 1.58 | 0.92 | 0.59 |
| IK9350-6 | 5 | 75.02 | 136.03 | 31 | 295 | 1.13 | 0.82 | 0.72 |
| IK9373A-6 | 6 | 75.81 | 134.58 | 46 | 322 | 1.46 | 0.93 | 0.64 |
| PS2728-2a-1 | 7 | 77.25 | 135.01 | 44 | 471 | 1.42 | 1.09 | 0.76 |
| PS2458-4* | 8 | 78.17 | 133.38 | 983 | 518 | 1.28 | 1.95 | 1.77 |

*For core PS2458-4, the $^9$B, $^{10}$Be and $^{10}$Be/$^9$Be results from the 30 cm sample are used as the core-top values.






In order to use $^{10}Be/^9Be$ as a synchronization tool, we must remove this influence of mixing riverine and marine
endmembers. It is non-trivial to derive a quantitative end-member mixing model solely from local sea-level
reconstructions because sea-level only provides conceptual evidence about the variable proportions of open ocean
and riverine water masses bathing the core site. Hence, we chose a statistical model, assuming that the changes in
the endmember-mixing were gradual and hence, can be removed by normalizing to the long-term trend in the
$^{10}Be/^9Be$ record. The residual centennial variability in $^{10}Be/^9Be$ is hypothesized to be driven by $^{10}Be$-production
rate changes and therefore suitable for synchronization.

Three different statistical models were used to test the sensitivity of our results to the choice of detrending
techniques. Figure 4a illustrates the three different trend fitting techniques (logarithmic, power, and LOESS
(LOcally Estimated Scatterplot Smoothing) applied to the $^{10}Be/^9Be$ data set. The relative $^{10}Be/^9Be$ residuals are
plotted with respect to the logarithmic, power and LOESS trends (Fig. 4b) and the differences fall within the
measurement uncertainties of the individual data points, showing that variations of the $^{10}Be/^9Be$ ratio are robust
against the choice of the detrending model.

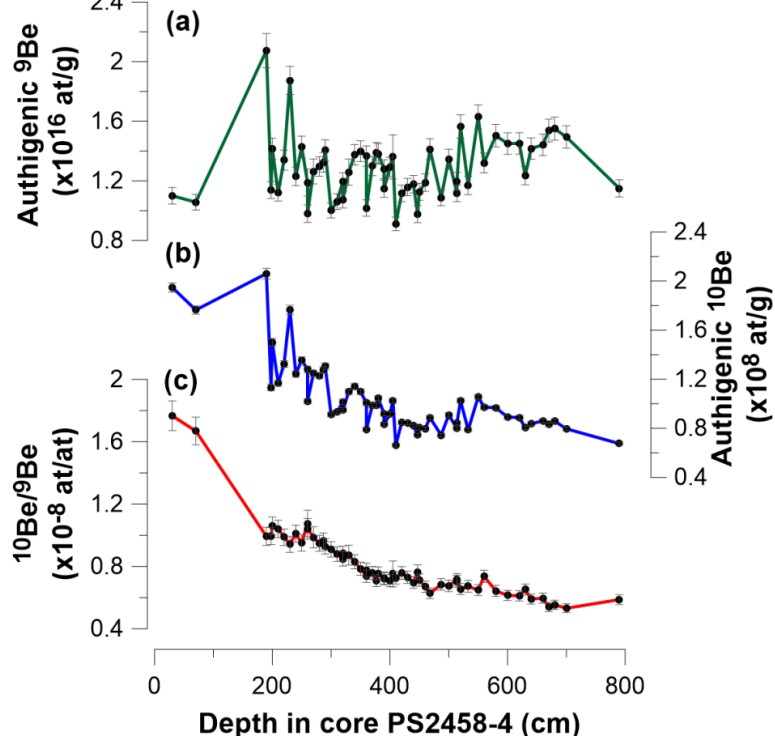


**Figure 3: Concentrations of (a) $^9Be$, (b) $^{10}Be$ and (c) $^{10}Be/^9Be$ atomic ratios from core PS2458-4**




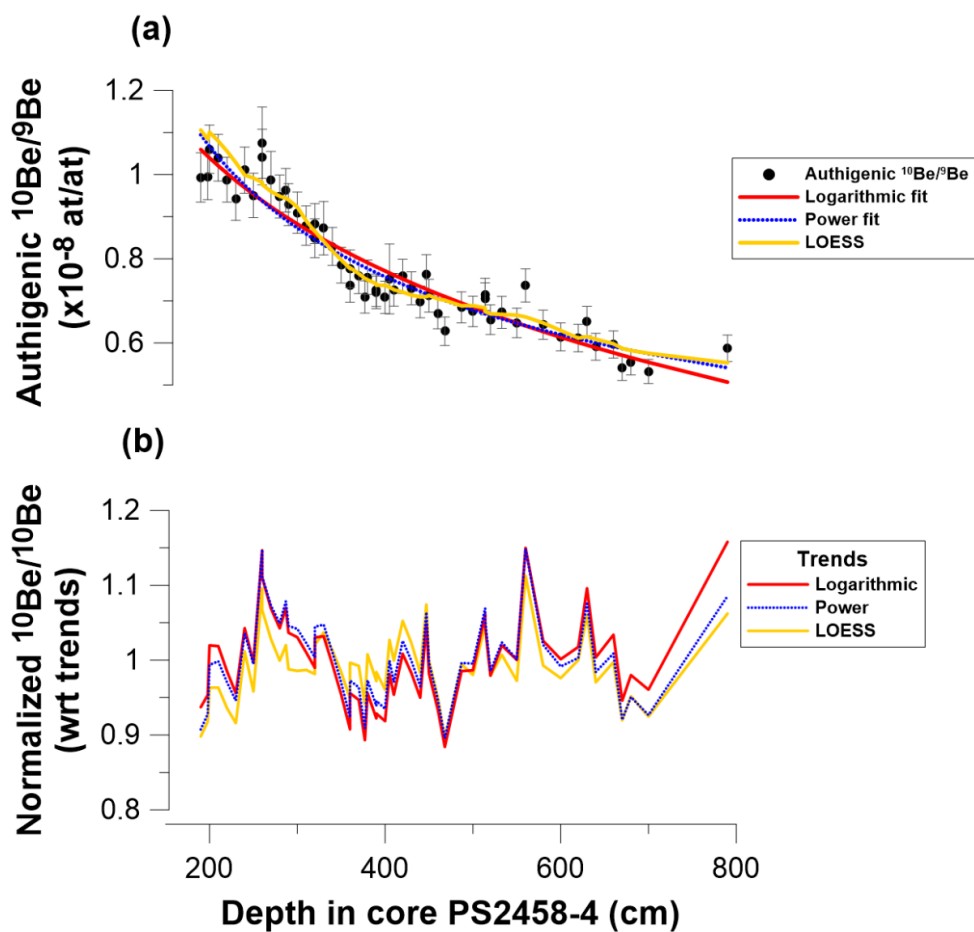

**Figure 4: Sensitivity tests (a) Three different trend fitting techniques (logarithmic, power, and LOESS), (b) Relative $^{10}Be/^{9}Be$ residuals with respect to logarithmic, power and LOESS trends**

To check whether the detrended $^{10}Be/^{9}Be$ record is driven by cosmogenic $^{10}Be$ production rate changes, we compare the detrended signal to the ice core $^{10}Be$-record. Figure 5 shows the ice core $^{10}Be$ record and PS2458-4 mean profile of the three detrended data sets with a 3-point LOESS graph plotted on an initial $^{14}C$-based age-scale (see used ΔR value below). Note however, that the following analyses have been performed on all three versions of the detrended dataset in order to test the robustness of our results against the choice of the detrending method. The variations observed in the sediment $^{10}Be/^{9}Be$ record follow closely the same pattern and relative amplitudes compared with the ice core $^{10}Be$ record. Therefore, we suggest that the variations observed in the $^{10}Be/^{9}Be$ record indeed reflect the production rate changes in the centennial range.

In order to refine the initial $^{14}C$-based chronology and infer a regional deglacial ΔR-estimate, we constructed $^{14}C$-based age-depth-models for PS2458-4 using OxCal 4.4 (Ramsey, 2009) assuming a range of ΔR between -110



(Bauch et al., 2001) and +800 [14]C years. Each age-model was then evaluated by comparing the resulting PS2458-
4 [10]Be/[9]Be-timeseries to the ice core [10]Be-record. For this purpose, we use the generalized likelihood function by
Christen and Pérez, (2009) that is otherwise used for the calibration of [14]C-dates:

$$L_{\Delta R} \propto \prod_{j=1}^{n} \left[ b + \frac{(x_j - y(t_j))^2}{2(\sigma_x^2 + \sigma_y^2)} \right]^{-(a+\frac{1}{2})}$$


In our case, the ice core provides the calibration that describes [10]Be-anomalies at each point in time (y(t)) which
is compared to the sediment [10]Be/[9]Be ($x_j$) on their modelled absolute age assuming a certain reservoir age. We use
a = 3 and b = 4 based on the recommendation of Christen and Pérez (2009). This allows us to use [10]Be to compare
the likelihoods of different age models, and thus [14]C-reservoir ages.

The likelihood values were calculated for each of the three different trend fitting techniques and are shown in
Figure 6. They result in a mean ΔR ± 1σ of 360 ± 75, 340 ± 50 and 335 ± 55 [14]C years for the logarithmic, power
and LOESS trend fitting techniques, respectively. These values are statistically indistinguishable and hence, we
opt for the arithmetic mean ΔR value of 345 ± 60 [14]C years. By using a global average marine reservoir age of
503 ± 63 [14]C years for the period 7.51-14.21 kyr BP (Heaton et al., 2020), we estimated a local MRA of 848 ± 90
[14]C years for the Laptev Sea during the deglaciation. The age-depth model for core PS2458-4 was reconstructed
using radiocarbon dates of mixed benthic bivalves and benthic foraminifera (Spielhagen et al., 2005). Therefore,
our calculated ΔR and corresponding MRA reflects to a benthic value.

The depositional age-depth model with a ΔR value of 345 ± 60 [14]C years for core PS2458-4 is shown in Figure
S2 in the Supplement accompanying this manuscript. Compared to the mean modelled ages calculated with a ΔR
value of -110 ± 28 [14]C years, the new modelled ages computed with a ΔR value of 345 ± 60 [14]C years were
observed to shift younger in the range of 429 to 707 years (Table S1).

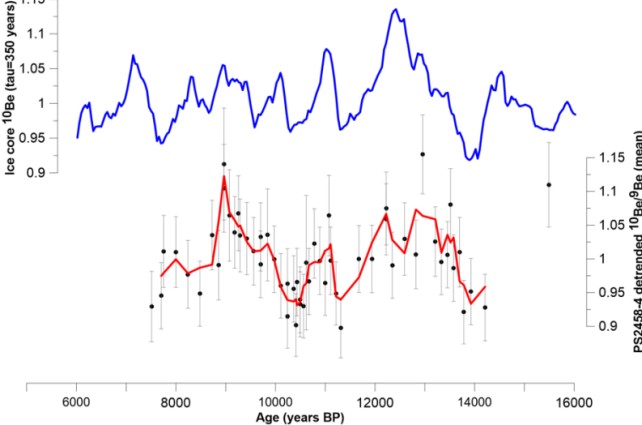


**Figure 5: Ice core [10]Be record with tau=350 years (blue) and PS2458-4 record calculated from the mean of the three**
**detrended data sets with a 3-point LOESS graph using ΔR value of 345±60 [14]C years for age-model (red)**




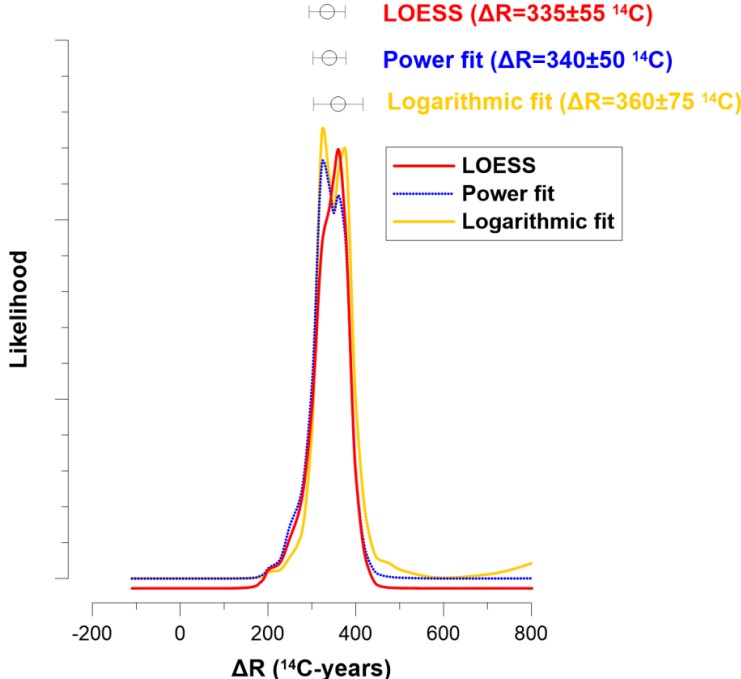

**Figure 6: Likelihood results with mean ΔR±1σ values of 360 ± 75, 340 ± 50 and 335 ± 55 ¹⁴C years BP based on**
**LOESS (red), power (blue dotted) and logarithmic (yellow) trend fitting techniques respectively.**

**Table 2. Radiocarbon and modelled ages from foraminifera and bivalve samples from core PS2458-4**

| Depth (cm) | Sample ID | ¹⁴C Age (¹⁴C years) | ± (years) | Modelled Age (mean) (cal BP) | Modelled Age (cal BP, 2σ) | Sample type |
|---|---|---|---|---|---|---|
| 667 | KIA6113 | 12600 | 110 | 13745 | 14089 – 13360 | mixed bivalves, benthic forams |
| 578 | AAR-3087 | 12270 | 65 | 13198 | 13428 – 12982 | mixed benthic forams |
| 530 | AAR-3086 | 11560 | 100 | 12551 | 12815 – 12244 | mixed benthic forams |
| 491* | AWI-7415.1.1 | 10968 | 159 | 11753 | 12220 – 11280 | mixed benthic forams |
| 467 | AAR-3085 | 10600 | 75 | 11291 | 11630 – 11005 | mixed benthic forams |
| 399 | AAR-3084 | 10090 | 65 | 10551 | 10811 – 10276 | mixed benthic forams |
| 369 | AAR-3083 | 10020 | 70 | 10357 | 10606 – 10135 | mixed benthic forams |
| 331.5* | AWI-7412.1.1 | 9596 | 122 | 9860 | 10183 – 9527 | mixed benthic forams |
| 291.5* | AWI-7411.1.1 | 9089 | 224 | 9305 | 9711 – 8917 | mixed benthic forams |
| 252 | AAR-3082 | 8830 | 55 | 8880 | 9129 – 8615 | mixed benthic forams |
| 241.5* | AWI-7410.1.1 | 8762 | 141 | 8762 | 9058 – 8448 | mixed benthic forams |
| 141.5* | AWI-7409.1.1 | 6447 | 158 | 6334 | 6696 – 5969 | mixed benthic forams |
| 121.5* | AWI-7408.1.1 | 6029 | 134 | 5985 | 6297 – 5638 | mixed benthic forams |
| 0.5* | AWI-7786.3.1 | 0 | | 0 | | mixed benthic forams |

Modelled ages were calculated using OxCal4.4 (Ramsey, 2009) with a ΔR value of 345±60 ¹⁴C years BP, as calculated in this study. Marine ¹⁴C dates were calibrated with the Marine20 curve (Heaton et al., 2020). The depth values with asterisks represent the new benthic foraminifera samples measured for ¹⁴C dates. The depth values without asterisks show the ¹⁴C dates published ¹⁴C dates from Spielhagen et al. (2005).





**4 Discussion**


We have been able to quantitatively compare the agreement between ice core [10]Be and sediment [10]Be/[9]Be for
different ΔR values and visually, we can observe how the two records representing cosmogenic radionuclide
production variations are in-phase with each other. It is a more robust approach to compare whole timeseries by
using a statistical method such as the likelihood function, instead of matching single wiggles with each other from
both records. The latter method is more prone to noise in each dataset and complicates the correct identification
of matching peaks.

When modelling the ice core data, we have assumed a 350-year residence time of [10]Be in the water column prior
to deposition. We tested the influence of choosing different residence times of [10]Be in the water column when
modelling the ice core data and then synchronizing the modeled data sets with the PS2458-4 [10]Be/[9]Be-timeseries.
Different tau values ($\tau = 200, 500, 600$ years) were used to model the ice core data and the ΔR-likelihood values
from the LOESS-smoothed [10]Be record were calculated. We observed that for all assumed tau-values likelihood
peaks occur at a ΔR value of 360 [14]C years (Fig. 7). This indicates that the most likely ΔR value is not strongly
dependent on the different assumed tau values. We found that only for the tau value of 200 years another best
likelihood estimate occurs at a ΔR value of 300 [14]C years BP, followed by the secondary likelihood maximum at
a ΔR value of 360 [14]C years BP. Figure S2 shows the modelled ice core time series with a tau value of 200 years,
which indicates clearly larger [10]Be amplitudes than what was calculated with a tau value of 350 years, which are
larger than the [10]Be/[9]Be changes seen in PS2458-4. Based on these results, it seems unlikely that the best likelihood
estimate occurring at a ΔR value of 300 [14]C years BP with tau=200 years is real.

Our calculated local benthic MRA value of $848 \pm 90$ [14]C years BP is consistent with the modern values calculated
by Bauch et al. (2001), which range from $295 \pm 45$ to $860 \pm 55$ [14]C years. The largest modern reservoir age of 860
$\pm 55$ [14]C years is located closest to the Lena Delta, which is comparable to the setting of the location of core
PS2458-4 during deglaciation around 14 - 12 kyr BP. Another study from the central Arctic Ocean reported MRA
values of 1400 [14]C years BP (ΔR = 1000) during the Late Glacial and 700 [14]C years BP (ΔR = 300) during the
Holocene (Hanslik et al., 2010).

The ΔR value was calculated during the deglaciation (14-8 kyr BP) and during this period the mean relative sea
level rose by about 64 m (Klemann et al., 2015). The core was retrieved at a depth of 983 m in 1994 and at 14 and
8 kyr BP the depths were about 903 and 967 m respectively. Moreover, as shown in Figure 1, the modern surface
[10]Be/[9]Be values show an increasing trend from the Lena Delta to the open ocean (Fig. 1). Thus, we attribute the
trend in PS2458-4 [10]Be/[9]Be to deglacial sea level rise and the associated coastline retreat (Bauch et al., 2001;
Klemann et al., 2015). During the glacial period, the core site was located close to the Lena River mouth and
hence, bathed in river-water with low [10]Be/[9]Be. With increasing sea-level and coastline retreat, open ocean waters
with higher [10]Be/[9]Be became more dominant.






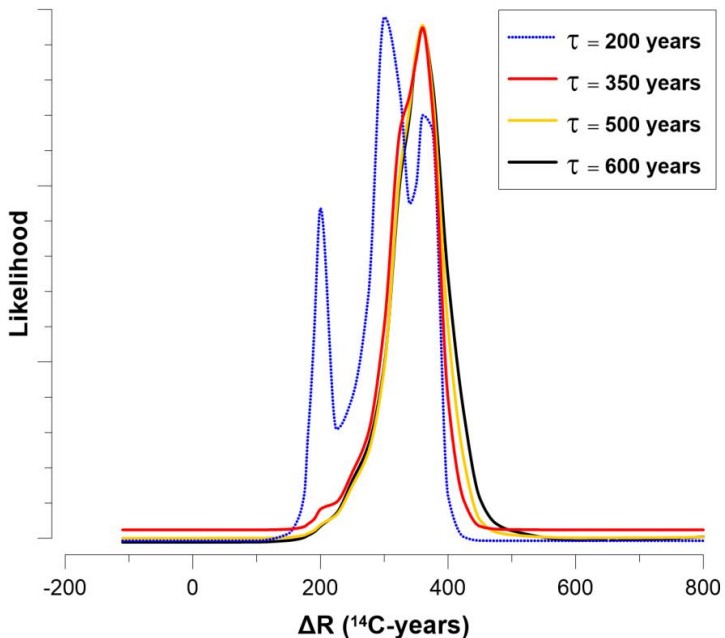


**Figure 7. Likelihood results based on different ΔR for the LOESS-smoothed ice core $^{10}$Be using for different tau values**
**of 200, 350, 500 and 600 years.**



We compared our estimated ΔR value 345 ± 60 $^{14}$C years with the approach proposed by Heaton et al. (2023) to
infer glacial ΔR values in polar regions. In the polar regions (outside 40° S - 40° N), it is expected that during
glacial episodes, there may have been regional differences in the amount of oceanic $^{14}$C depletion compared to
the global non-polar ocean mean represented by Marine20. The increase in the volume and density of sea ice
limiting air-sea gas-exchange may cause a significant larger ΔR during the glacial era compared to the interglacial
values. For glacial periods (55.0 - 11.5 kyr BP), Heaton et al. (2023) proposed a latitude-dependent method to
infer upper bounds of the possible ΔR difference between Holocene and Glacial in polar regions. A lower bound
$ΔR^{Hol}$ is based on samples from the Holocene and an upper (glacial) bound $ΔR^{GS}$, is calculated by increasing $ΔR^{Hol}$
depending on the latitude.

The PS2458-4 record used in this study extends from about 7.5 to 14.2 kyr BP and therefore covers the early
Holocene and parts of the deglacial period. Thus, from 11.5 to 14.2 kyr BP, the record extends into the glacial and
samples from this period may require a glacial polar boost as proposed by Heaton et al. (2023). We calculated
$ΔR^{Hol}$ from $^{14}$C samples found in the online database at http://calib.org/marine/ (Reimer and Reimer, 2001). Using
the weighed mean value of the 5 nearest ΔR values from the core location in the Laptev Sea from Bauch et al.
(2001), yields a $ΔR^{Hol}$ value of -95 ± 61 $^{14}$C years. $ΔR^{GS}$ was calculated as: $ΔR^{GS} = ΔR^{Hol} + ΔR^{Hol→GS}$, in agreement
with the GS scenario as described in Heaton et al. (2023). The value $ΔR^{Hol→GS}$ is dependent on the latitude of the





sample and at 78.75 °N, it amounts to 790 $^{14}$C years. The resulting $\Delta R^{GS}$ value is 695 ± 61 $^{14}$C years and is much
larger than our inferred benthic ΔR value (345 ± 60 $^{14}$C years).

These differences are likely due to distinct regional changes in climate and hydrology. At the core location in the
Laptev Sea, sea-ice cover was less during the Younger Dryas and Heinrich Stadial 1 compared to the Holocene
(Fahl and Stein, 2012), contrary to large-scale deglacial sea ice trends included in the model by Heaton et al.
(2023). The expansion of regional sea-ice cover during the recent past in the Laptev Sea could have further
influenced the ΔR value, which then should have been larger during the Holocene compared to the early
deglaciation. However, our calculated ΔR value of 345 ± 60 $^{14}$C years is larger than the modern average ΔR value
of -95 ± 61 $^{14}$C years, making it unlikely that sea-ice cover dynamics were the main driver of past changes in
regional ΔR. Instead, as mentioned before, the local reservoir ages in the region are spatially highly variable and
influenced by a hardwater effect (Bauch et al. 2001). These regional processes are thus site specific and hence,
obviously cannot be covered by the approach of Heaton et al. (2023).  Bauch et al. (2001) reported that the
relatively old $^{14}$C-age of bivalve shells collected in proximity of the Lena Delta near Tiksi Bay, might be due to
the influence of local hardwater effect. This is consistent with the modern setting where the largest ΔR is found
close to the Lena Delta and lower ΔR towards the shelf-edge  (Bauch et al., 2001). Hence, the larger deglacial ΔR
of PS2458-4 could be driven by its proximity to the Lena River during that time as evidenced by low $^{10}$Be/$^9$Be as
discussed earlier.


**5 Conclusion**
We present high-resolution $^9$Be and $^{10}$Be records reconstructed from core PS2458-4, which was retrieved from the
continental slope of the eastern Laptev Sea in the Arctic Ocean. We demonstrate that these records are influenced
by the distance of the core site to the Lena River, which changed depending on sea-level. Centennial to millennial
scale variability in the $^{10}$Be/$^9$Be ratio can be attributed to variations in production rate and can hence be used to
correlate our sediment record to ice-core $^{10}$Be records.

This is the first study to reconstruct high-resolution $^{10}$Be production rate changes from $^{10}$Be/$^9$Be records from
Arctic marine sediments for correlation to ice cores, and this approach has been applied with success. We have
correlated the $^{10}$Be from marine sediment core PS2458-4 with $^{10}$Be from ice core and used a likelihood function
to estimate ΔR values.

Our estimate for the deglacial benthic ΔR value for the Laptev Sea is 345 ± 60 $^{14}$C years BP corresponding to a
MRA of 848 ± 90 $^{14}$C years. The ΔR value will be used to refine the age-depth model for core PS2458-4 from the
Laptev Sea, which could be used as a reference chronology for the Laptev Sea.

**Data availability**

The $^9$Be, $^{10}$Be and $^{10}$Be/$^9$Be data sets from core PS2458-4 generated in this study are available as a Supplement
to this paper.



**Author contributions**

FA and GM designed the study. AN, MM conducted the laboratory analyses and FA, AN and GM analyzed the data. JL and KS were responsible for preparation and conduction of the $^{10}$Be AMS measurements. JW selected appropriate foraminifera samples for radiocarbon dating. HG undertook the radiocarbon measurement of the foraminifera samples and analyzed the data. AN drafted a first version of the paper and FA and AN generated the figures. All co-authors contributed to the writing and provided feedback on the paper.

**Competing interests**

The contact author has declared that neither of the authors has any competing interests.

**Acknowledgements**

Parts of this research were carried out at the Ion Beam Centre (IBC) at the Helmholtz-Zentrum Dresden-Rossendorf e. V., a member of the Helmholtz Association. We would like to thank the DREAMS operator team for their assistance with AMS-measurements. FA was supported by the Helmholtz Association (VH-NG 1501). We are grateful for the technical support offered by Torben Gentz and Elizabeth Bonk from the MICADAS facility at AWI Bremerhaven. AN would like to thank DAAD and POLMAR for support during his doctoral studies.



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
