# Peer review of "Precise dating of deglacial Laptev Sea sediments via 14C and authigenic 10Be/9Be – assessing local 14C reservoir ages"

_EGUsphere, 2024_

## Author Comment (AC1)

**Response to comments by reviewer 1**

The authors would like to thank the reviewer 1 for the valuable comments and the suggested modifications/ improvements, which will help us to enhance the content of the manuscript.

Hereunder, we address the comments and suggestions from the reviewer 1 and provide the response to each point raised. Our responses are given in black colour.

**Reviewer comment:**

**ΔR instead of R**: The focus of the paper shifts from Marine Reservoir Age (MRA) R to ΔR (line 60-63) to discuss local deviations from a modelled global MRA that is, however, uncertain at high latitudes. The definition of MRA is the deviation from the atmosphere (globally well mixed, line 47-48). Since the correlation of the sediment core with the $^{10}Be/^9Be$ ice core record provides an independent time scale and thus access to the IntCal20 atmospheric $^{14}C$ record, the local MRA at any place in the sediment core record follows from a simple comparison between the measured sediment $^{14}C$ concentration and IntCal20.

Reply:
Thank you for this thoughtful comment. The framework of our study is designed to estimate the likelihood of ΔR, rather than the likelihood of R directly. The reason is, that the R-estimate in Marine20 include some aspects that will also affect high-latitude records, such as the dependence of air-sea gas exchange on $CO_2$-levels and transient changes of atmospheric $\Delta^{14}C$. Of course, additional factors can affect local R especially at high latitudes. However, we want to point out that when applying a constant ΔR we obtain a good for match for $^{10}Be$ and hence, we don't see any evidence for a variable ΔR from our data.

**Reviewer comment:**

**Local ΔR range**: Comparison with Heaton et al. 2023 modelled Laptev Sea MRA will give local ΔR values for this location near the Lena mouth at the edge of the continental shelf under changing sea level and climate that can be compared with the range of values (-100 to +800 yr) mentioned.

Reply:
Yes correct, in this study we are comparing our estimated ΔR value with the modern ΔR values. We discuss this topic in the Discussion section in the paragraph from line 362 to 370.

**Reviewer comment:**

**Figure 2**: Figure 2 provides $^{10}Be/^9Be$ after climate correction. These assumed corrections are based on our best understanding of the $^{10}Be$ production, its distribution, and local snow accumulation but the corrections may have flaws. It is thus interesting to compare the $^{10}Be/^9Be$ record of Figure 2 with the NORTHGRIP $\delta^{18}O$ climate record and the IntCal20 $\Delta^{14}C$ record (supplemental figure).

From 6000 to ~14500 yr BP there is detailed agreement between WAIS and GISP2 while beyond this, to 18000 yr BP the agreement is worse. This change in character closely coincides

with the resumption of a strong Atlantic Meridional Overturning Circulation (AMOC) and the start of the Bølling in Greenland. IntCal20 shows a steep decrease in $\Delta^{14}$C slightly earlier, more coeval with the apparently opposite excursions in $^{10}$Be/$^9$Be in WAIS and GISP2. Is this a problem of the Polar Seesaw? The discussion of one high-latitude MRA and one $\Delta$R thus does not do justice to the data presented.

Comparison of the three records further indicates coincidence of the Older Dryas climate episode around 14000 yr BP with a $^{10}$Be/$^9$Be low and a little increase in $\Delta^{14}$C. (AMOC?) The end of the Allerød/start Younger Dryas shows again coeval cooling and $\Delta^{14}$C increase with a possible $^{10}$Be/$^9$Be low but here the rapid changes in the $^{10}$Be/$^9$Be record around this time require a more detailed synchronisation to be substantiated. A challenge to the authors.

Reply:

We appreciate your insightful suggestion. The MRA is changing over this time period and it is only the $\Delta$R that is constant. In the Marine20, modelled MRA-changes do not account for changes in AMOC, however, the MRA mainly depends on the equilibration between atmosphere and surface ocean and hence, air-sea gas exchange. So as long as atmospheric $\Delta^{14}$C is prescribed in the model, the MRA estimate should be reliable (see Köhler et al. 2024). Further, the Laptev Sea is a shallow shelf sea in the tightly enclosed Arctic Ocean with limited exchange with the North Atlantic. It is hence, not clear whether an AMOC-imprint can be expected in this setting. Indeed, climate records from PS2458-4 do not show any evidence for such a coupling during the deglaciation (Spielhagen et al., 2005). Instead, our analysis shows that using a single, constant $\Delta$R value provides a robust match for the Beryllium records. Therefore, we believe that our current methodology is both adequate and appropriate.

Indeed, we recognize the discrepancies between the WAIS and GISP2 records in the period spanning approximately 14,500 to 18,000 years BP. These differences may be due to a variety of reasons: i) The Greenland ice core timescale shows large biases during this period (Adolphi et al. 2018) which would affect the inferred snow accumulation rates and timing of $^{10}$Be oscillations, ii) changes in the importance of wet and dry deposition of $^{10}$Be into the ice, iii) aerosol transport to Greenland and Antarctica. It is clearly beyond the scope of the paper to discuss or attempt to correct for these effects. Instead, our analysis has deliberately focused on the common structures evident in both records. It is important to note that this particular time frame older than 14,500 years BP is not central to our synchronization efforts. Our synchronization methodology primarily utilizes the period from 6,000 to approximately 14,500 years BP, where there is strong agreement between the WAIS and GISP2 records. This approach allows us to establish a more robust and reliable synchronization. Regarding the changes in $\Delta^{14}$C and the potential influence of the AMOC, while undoubtedly intriguing, we believe that a comprehensive examination of these factors falls outside the primary scope of our current study.

To elucidate the variability in our records during the Older Dryas climate event (~14,000 years BP) and other periods, we have generated a comprehensive graph (see Fig. A1 below). Recent research by Bard et al. (2023) suggests that the observed excursion in $\Delta^{14}$C values around this period is attributable to a century-long cosmogenic overproduction event, analogous to the Maunder-type solar minima frequently observed in the past millennia. While a comparative

analysis of $^{10}Be/^9Be$ record variations with $\Delta^{14}C$ during both the Older Dryas and Younger Dryas periods would be informative, such an investigation extends beyond the primary scope of this study.

[Figure]

Figure A1. (a) Atmospheric $CO_2$ (Köhler et al., 2017). (b) $\Delta^{14}C$ (Reimer et al., 2020). (c) NGRIP $\delta^{18}O$ (Andersen et al., 2004). (d) Rate of global sea-level change (Lambeck et al., 2014). (e) PS2458-4 record calculated from the mean of the three detrended data sets with a 3-point LOESS graph using $\Delta R$ value of 345±60 $^{14}C$ years for age-model. (f) Ice core $^{10}Be$ record with tau=350 years. (g) WAIS (Muschitiello et al., 2019; Sigl et al., 2016; Sinnl et al., 2023). (h) GISP2 *(Finkel & Nishiizumi, 1997)*. The pin marks at the bottom represent the age control points of core PS2458-4.

**Reviewer comment:**

**Table S1:** The uncertainty intervals of several of the dated samples straddle climate changes in Greenland. The known timing of these changes may, in combination with the sample position in the sediment isotope/climate record, be used to refine the dating intervals. In this respect it would be good to know more about the mixed benthic. Is there information about endobenthic versus epibenthic contributions?

Reply:

Thank you for this suggestion. The measured geochemical parameters from our sediment core does not give a clear indication of the Older Dryas or the Younger Dryas climate signals that can be related to climate changes in Greenland.

More information about the mixed benthic foraminifera and mixed bivalves will be added in Section 2.1 and species details will be included in Table S1 in the revised manuscript (see revised Table S1 hereunder).

**Table S1. Radiocarbon and modelled ages from foraminifera and bivalve samples from core PS2458-4**

| Depth (cm) | $^{14}C$ Age ($^{14}C$ years) | ± (years) | ($\Delta R= 345 \pm 60$ $^{14}C$ years BP) | | ($\Delta R= -110 \pm 28$ $^{14}C$ years BP) | | Modelled Age (difference) (cal BP) | Sample type | Species |
|---|---|---|---|---|---|---|---|---|---|
| | | | Modelled Age (mean) (cal BP) | Modelled Age (cal BP, 2σ) | Modelled Age (mean) (cal BP) | Modelled Age (cal BP, 2σ) | | | |
| 667 | 12600 | 110 | 13745 | 14089 – 13360 | 14452 | 14870 – 14009 | 707 | mb, mbf | *Thyasira* sp., *Yoldiella* sp. |
| 578 | 12270 | 65 | 13198 | 13428 – 12982 | 13687 | 13931 – 13470 | 489 | mb | *Thyasira* sp., *Yoldiella* sp. |
| 530 | 11560 | 100 | 12551 | 12815 – 12244 | 12980 | 13199 – 12748 | 429 | mb | *Thyasira* sp., *Yoldiella* sp. |
| 491* | 10968 | 159 | 11753 | 12220 – 11280 | 12371 | 12692 – 12026 | 618 | mbf | *L. lobatula, C. neoteretis* |
| 467 | 10600 | 75 | 11291 | 11630 – 11005 | 11973 | 12279 – 11683 | 682 | mb | *Thyasira* sp., *Yoldiella* sp. |
| 399 | 10090 | 65 | 10551 | 10811 – 10276 | 11185 | 11397 – 10991 | 634 | mb | *Thyasira* sp., *Yoldiella* sp. |
| 369 | 10020 | 70 | 10357 | 10606 – 10135 | 10966 | 11187 – 10746 | 609 | mb | *Thyasira* sp., *Yoldiella* sp. |
| 331.5* | 9596 | 122 | 9860 | 10183 – 9527 | 10456 | 10757 – 10172 | 596 | mbf | *I. helenae, I. norcrossi, C. neoteretis* |
| 291.5* | 9089 | 224 | 9305 | 9711 – 8917 | 9890 | 10230 – 9529 | 585 | mbf | *C. neoteretis* |
| 252 | 8830 | 55 | 8880 | 9129 – 8615 | 9432 | 9594 – 9258 | 552 | mb | *Thyasira* sp., *Yoldiella* sp. |
| 241.5* | 8762 | 141 | 8762 | 9058 – 8448 | 9310 | 9527 – 9044 | 548 | mbf | *I. helenae, I. norcrossi, C. neoteretis* |
| 141.5* | 6447 | 158 | 6334 | 6696 – 5969 | 6838 | 7177 – 6489 | 504 | mbf | *C. neoteretis* |
| 121.5* | 6029 | 134 | 5985 | 6297 – 5638 | 6463 | 6790 – 6143 | 478 | mbf | *C. neoteretis* |
| 0.5* | 0 | | 0 | | | | | mbf | *C. lobatulus* |

Modelled ages were calculated using OxCal4.4 (Ramsey, 2009) with corresponding ΔR values. Marine $^{14}C$ dates were calibrated with the Marine20 curve (Heaton et al., 2020). The depth values with asterisks represent the new benthic foraminifera samples measured for $^{14}C$ dates. The depth values without asterisks show the $^{14}C$ dates published from (Spielhagen et al., 2005). Libby half-life (5568 years) was used to calculate $^{14}C$ age of foraminifera samples. The modelled age (difference) is calculated by subtracting the modelled age (mean) with ΔR= -110 ± 28 $^{14}C$ years BP from the modelled age (mean) with ΔR= 345 ± 60 $^{14}C$ years BP. Sample type: mb= mixed bivalves, mbf= mixed benthic foraminifera.

Paragraph to include in Section 2.1: The mixed bivalve species used in Spielhagen et al. (2005) were described as *Thyasira* sp. and *Yoldiella* sp. Both bivalve species typically occur in cold water environments at continental margins and in areas of limited food supply, as is the Laptev Sea continental margin. Concerning the mixed benthic foraminifera species, usually epibenthic species such as *Lobatula lobatula* are preferred. Since this latter species is rare in

our sediment samples, other species such as: *Cassidulina neoteretis*, *Islandiella helenae* and *Islandiella norcrossi* were selected for radiocarbon dating. In the Arctic Ocean all these species live close to the sediment surface (Wollenburg & Kuhnt, 2000; Wollenburg & Mackensen, 1998a, 1998b) and reflect the carbon and oxygen isotope record of the bottom water in their shells.

**Reviewer comment:**

**Half-life**: Line 66 gives the Audi et al., 2003 half-life of 5700 yr.  The value of 5730 yr. is still commonly used in reporting $^{14}$C results. Please state clearly what has been used in tables 2 and S1.

Reply:
Thank you for pointing this out. For the calculation of the 'radiocarbon age' of the foraminifera samples, the age is calculated using the Libby half-life of 5568 years, by using the following equation:  Age= - 8033 ln (F$^{14}$C). Even though it is technically incorrect, the Libby half-life remains the age-dating convention so as to avoid the confusion by attempting to update older literature.  In Table 2 and S1, the Libby half-life was used and we will add this information in the revised manuscript at the bottom of the Tables.

**Reviewer comment:**

**Discussion**: Robustness is important for calibration and can argue for a statistical use of an averaged MRA or ΔR value (lines 309-314). Yet, the time interval considered includes large climatic changes, changes in AMOC, and sea level change and, therefore, large changes in MRA and ΔR are to be expected.

 It will be good to build the discussion on the record of changing MRA values over time, obtained from the direct comparison of the synchronized PS2458-4 record with IntCal20. Using a moving time window of 1000 or 1500 years to calculate a 'temporal' best fit instead of the full period may give robustness and flexibility and show differences for the H1, the Bølling/Allerød, Younger Dryas, and (Pre)Boreal periods.

Reply:
Thank you for this suggestion. According to the initial radiocarbon-based age model that we used, the selected sediment samples covered three large cosmogenic radionuclide production rate swings, as evidenced by ice core $^{10}$Be and tree-ring $^{14}$C records (e.g., Adolphi & Muscheler, 2016), that occurred between 8,500 and 11,500 kyr BP. The variations that were observed in the sediment $^{10}$Be/$^{9}$Be record follow closely the same pattern and relative amplitudes compared with the ice core $^{10}$Be record (Fig. 5). Therefore, it was suggested that the variations observed in the $^{10}$Be/$^{9}$Be record indeed reflect the production rate changes in the centennial range. As we discussed, we believe that it is a more robust approach to compare whole timeseries by using a statistical method such as the likelihood function (after removing additional trends from influence of mixing riverine and marine endmembers), instead of matching single wiggles (or shorter time periods of 1,000 years) with each other from both records. The latter method is more prone to noise in each dataset and complicates the correct

identification of matching peaks. Moreover, using just a single ΔR, we found that there is a good match between the Be records. Hence this indirectly supports our assumption of a constant ΔR as we should otherwise have obtained a bad match. However, we want to point out, that a constant ΔR does not imply a constant MRA (which is variable in Marine20) but just a constant offset. Figure A2 below shows the $^{14}$C Age calculated from the foraminifera samples plotted together with Intcal20 and Marine20 curves. Figure A3 shows the non-polar global-average MRA corresponding to Marine20 and the inferred MRA calculated by subtracting the atmospheric $^{14}$C age (derived from Intcal20) from the $^{14}$C age of foraminifera and bivalves samples. Based on the results we observe that the inferred MRA data points follow closely the Marine20 MRA+ΔR data, indicating a good match. This is of course partly expected by design (i.e., through calibration with a constant ΔR), but the fact that this reconciles the $^{14}$C-based age model and the 10Be data from ice cores and sediments, gives us confidence in this result.

[Figure]

Figure A2. Foraminifera ages plotted with Marine 20 (Heaton et al., 2020) and Intcal20 (Reimer et al., 2020).

[Figure]

Figure A3. Non-polar global-average MRA corresponding to Marine20 (Heaton et al., 2020) added to ΔR value 345 $^{14}$C years (blue) and the inferred MRA calculated by subtracting the atmospheric $^{14}$C age (derived from Intcal20) from the $^{14}$C age of foraminifera and bivalves samples (orange).

**Reviewer comment:**

**Figure 4b**: The Y-axis should read $^{10}$Be/$^{9}$Be

Reply:
The Y-axis of Fig. 4b will be changed accordingly.

**References:**

Adolphi, F., & Muscheler, R. (2016). Synchronizing the Greenland ice core and radiocarbon timescales over the Holocene-Bayesian wiggle-matching of cosmogenic radionuclide records. *Climate of the Past*, *12*(1), 15–30. https://doi.org/10.5194/cp-12-15-2016

Andersen, K. K., Azuma, N., Barnola, J. M., Bigler, M., Biscaye, P., Caillon, N., Chappellaz, J., Clausen, H. B., Dahl-Jensen, D., Fischer, H., Flückiger, J., Fritzsche, D., Fujii, Y., Goto-Azuma, K., Grønvold, K., Gundestrup, N. S., Hansson, M., Huber, C., Hvidberg, C. S., … White, J. W. C. (2004). High-resolution record of Northern Hemisphere climate extending into the last interglacial period. *Nature*, *431*(7005). https://doi.org/10.1038/nature02805

Bard, E., Miramont, C., Capano, M., Guibal, F., Marschal, C., Rostek, F., Tuna, T., Fagault, Y., & Heaton, T. J. (2023). A radiocarbon spike at 14 300 cal yr BP in subfossil trees provides the impulse response function of the global carbon cycle during the Late Glacial. *Philosophical Transactions of the Royal Society A: Mathematical, Physical and Engineering Sciences*, *381*(2261). https://doi.org/10.1098/rsta.2022.0206

Finkel, R. C., & Nishiizumi, K. (1997). Beryllium 10 concentrations in the Greenland Ice Sheet Project 2 ice core from 3-40 ka. *Journal of Geophysical Research: Oceans*, *102*(C12). https://doi.org/10.1029/97JC01282

Heaton, T. J., Köhler, P., Butzin, M., Bard, E., Reimer, R. W., Austin, W. E. N., Bronk Ramsey, C., Grootes, P. M., Hughen, K. A., Kromer, B., Reimer, P. J., Adkins, J., Burke, A., Cook, M. S., Olsen, J., & Skinner, L. C. (2020). Marine20 - The Marine Radiocarbon Age Calibration Curve (0-55,000 cal BP). *Radiocarbon*, *62*(4). https://doi.org/10.1017/RDC.2020.68

Köhler, P., Nehrbass-Ahles, C., Schmitt, J., Stocker, T. F., & Fischer, H. (2017). A 156 kyr smoothed history of the atmospheric greenhouse gases CO2, CH4, and N2O and their radiative forcing. *Earth System Science Data*, *9*(1). https://doi.org/10.5194/essd-9-363-2017

Köhler, P., Skinner, L. C., & Adolphi, F. (2024). Simulated radiocarbon cycle revisited by considering the bipolar seesaw and benthic 14C data. *Earth and Planetary Science Letters*, *640*. https://doi.org/10.1016/j.epsl.2024.118801

Lambeck, K., Rouby, H., Purcell, A., Sun, Y., & Sambridge, M. (2014). Sea level and global ice volumes from the Last Glacial Maximum to the Holocene. *Proceedings of the National Academy of Sciences of the United States of America*, *111*(43). https://doi.org/10.1073/pnas.1411762111

Muschitiello, F., D'Andrea, W. J., Schmittner, A., Heaton, T. J., Balascio, N. L., deRoberts, N., Caffee, M. W., Woodruff, T. E., Welten, K. C., Skinner, L. C., Simon, M. H., & Dokken, T. M. (2019). Deep-water circulation changes lead North Atlantic climate during deglaciation. *Nature Communications*, *10*(1). https://doi.org/10.1038/s41467-019-09237-3

Ramsey, C. B. (2009). Bayesian analysis of radiocarbon dates. *Radiocarbon*, *51*(1). https://doi.org/10.1017/s0033822200033865

Reimer, P. J., Austin, W. E. N., Bard, E., Bayliss, A., Blackwell, P. G., Bronk Ramsey, C., Butzin, M., Cheng, H., Edwards, R. L., Friedrich, M., Grootes, P. M., Guilderson, T. P., Hajdas, I., Heaton, T. J., Hogg, A. G., Hughen, K. A., Kromer, B., Manning, S. W., Muscheler, R., … Talamo, S. (2020). The IntCal20 Northern Hemisphere Radiocarbon Age Calibration Curve (0-55 cal kBP). *Radiocarbon*, *62*(4). https://doi.org/10.1017/RDC.2020.41

Sigl, M., Fudge, T. J., Winstrup, M., Cole-Dai, J., Ferris, D., McConnell, J. R., Taylor, K. C., Welten, K. C., Woodruff, T. E., Adolphi, F., Bisiaux, M., Brook, E. J., Buizert, C., Caffee, M.

W., Dunbar, N. W., Edwards, R., Geng, L., Iverson, N., Koffman, B., … Sowers, T. A. (2016). The WAIS Divide deep ice core WD2014 chronology – Part 2: Annual-layer counting (0–31 ka BP). *Climate of the Past*, *12*(3), 769–786. https://doi.org/10.5194/cp-12-769-2016

Sinnl, G., Adolphi, F., Christl, M., Welten, K. C., Woodruff, T., Caffee, M., Svensson, A., Muscheler, R., & Rasmussen, S. O. (2023). Synchronizing ice-core and U/Th timescales in the Last Glacial Maximum using Hulu Cave 14C and new 10Be measurements from Greenland and Antarctica. *Climate of the Past*, *19*(6). https://doi.org/10.5194/cp-19-1153-2023

Spielhagen, R. F., Erlenkeuser, H., & Siegert, C. (2005). History of freshwater runoff across the Laptev Sea (Arctic) during the last deglaciation. *Global and Planetary Change*, *48*(1-3 SPEC. ISS.), 187–207. https://doi.org/10.1016/j.gloplacha.2004.12.013

Wollenburg, J. E., & Kuhnt, W. (2000). The response of benthic foraminifers to carbon flux and primary production in the Arctic Ocean. *Marine Micropaleontology*, *40*(3). https://doi.org/10.1016/S0377-8398(00)00039-6

Wollenburg, J. E., & Mackensen, A. (1998a). Living benthic foraminifers from the central Arctic Ocean: Faunal composition, standing stock and diversity. *Marine Micropaleontology*, *34*(3–4). https://doi.org/10.1016/S0377-8398(98)00007-3

Wollenburg, J. E., & Mackensen, A. (1998b). On the vertical distribution of living (Rose Bengal stained) benthic foraminifers in the Arctic Ocean. *Journal of Foraminiferal Research*, *28*(4). https://doi.org/10.2113/gsjfr.28.4.268

---

## Author Comment (AC2)

**Response to comments by reviewer 2**

The authors thank the anonymous reviewer 2 for the valuable comments on the manuscript. We have carefully taken note of the comments and will make the necessary revisions to address the suggestions. Our responses are given in black colour.

**Reviewer comment:**

I think the marine reservoir age discussion has to be clarified. Usually the marine reservoir age refers to the 14C age difference between upper ocean (mixed layer) and the atmosphere. However, this study discusses benthic 14C ages at a present depth of about 1000 m (less during the deglaciation). The setting is not an open ocean setting and, therefore, I assume that the authors have good reasons to relate their 14C offset to Marine20. However, this is not at all explained and should be discussed thoroughly.

Reply:
Thank you for this comment. We agree that usually the marine reservoir age refers to the $^{14}$C age difference between upper ocean (mixed layer) and the atmosphere. However, in our study, we used radiocarbon ages of mixed benthic bivalves and mixed benthic foraminifera to construct our initial age-depth model using ΔR value of $-110\pm28$ $^{14}$C years BP. The benthic bivalves and foraminifera species live close to the sediment surfaces and reflect the carbon and oxygen isotope record of the bottom water in their shells. Therefore, they are recording deep water signals and we relate our calculated deglacial ΔR value to be a benthic value. We agree that Marine20 provides a surface MRA, however, within the first 1000 m of the water column, $\Delta^{14}$C gradients are still relatively small and especially changes in the MRA, which are set at the surface, will be comparable. Offsets on the other hand, are included through the application of a ΔR. A paragraph has been added in the discussion section in the revised manuscript to clarify these points.

**Reviewer comment:**

If I understood correctly, the authors assume a constant reservoir effect in their calculations. Is there any discernible trend in the reservoir age over the deglaciation and wouldn't one expect a trend considering the changing setting (affecting so strongly 10Be/9Be).

Reply:
Thank you for this suggestion. From our data, we cannot robustly infer a trend in the reservoir ages (See also reply to Reviewer 1), but we also cannot rule this out completely with the method that we employed. However, we note that applying a constant ΔR leads to a good agreement between the 10Be-records, thus not providing any evidence for a time-variable ΔR. Furthermore, we believe that we cannot match single 10Be-wiggles as the noise in the data is quite high. We think that our conservative approach best serves the reliability of our findings.

**Reviewer comment:**

The authigenic 10Be/9Be record is dominated by a large trend and the residual variability appears to be largely within the measurement uncertainties (see e.g. Fig 4a where few points deviate from the trend lines exceeding their uncertainties). I recommend that the authors elaborate more if these deviations from the trend can be considered statistically significant.

Reply:

Thank you for this insightful suggestion. We took this into account and the measurement uncertainty are entering into the calculation of this match (see equation 1; more details were given in lines 247-252). We have elaborated on these deviations from the trends and we show that our results are robust against different detrending techniques. By jointly analyzing all samples, we achieve statistically significant results that support the reliability of our findings.

**Reviewer comment:**

The authors mention replicate measurements but do not seem to discuss them. I assume that they are shown in e.g. figure 4 but it could be discussed more (e.g. where the replicates separate samples from the same depth or e.g. replicate measurements on the same sample after leaching). To which extend do the replicates agree?

Reply:

Thank you for pointing this out. In Table S2 in the Supplement, 5 replicate samples are given (260, 320, 360, 390 and 514 cm) and their corresponding values are shown. Table A1 below shows the coefficient of variation results expressed in percentage (%) for each replicate. The agreement between replicate measurements of $^{10}$Be/$^{9}$Be ratios was assessed using the Coefficient of Variation (CV) for each depth. We observe that the authigenic $^{10}$Be/$^{9}$Be ratios demonstrated relatively low CV values, ranging from 0.98% to 7.11%, which is in agreement with the stated uncertainties of the $^{10}$Be/$^{9}$Be-ratio. The CV results and description has been added in the revised manuscript.

Table A1. Coefficient of variation values.

| Depth (cm) | Authigenic 10Be/9Be (at/at) [x10^-8] | sigma [%] | Authigenic 10Be/9Be Coefficient of Variation [%] |
|---|---|---|---|
| 260 | 1.08 | 7.87 | 7.11 |
| 260 | 1.04 | 6.34 | |
| 320 | 0.85 | 5.46 | 2.45 |
| 320 | 0.88 | 5.35 | |
| 360 | 0.74 | 5.43 | 3.72 |
| 360 | 0.78 | 5.75 | |
| 390 | 0.72 | 5.39 | 0.98 |
| 390 | 0.73 | 5.36 | |
| 514 | 0.72 | 5.37 | 5.39 |
| 514 | 0.70 | 5.40 | |